# DeepDive: Advancing Deep Search Agents with Knowledge Graphs and Multi-Turn RL

## Abstract

Augmenting large language models (LLMs) with browsing tools substantially improves their potential as deep search agents to solve complex, real-world tasks. Yet, open LLMs still perform poorly in such settings due to limited long-horizon reasoning capacity with browsing tools and the lack of sufficiently difficult supervised data. To address these challenges, we present DeepDive to advance deep search agents. First, we propose a strategy to automatically synthesize complex, difficult, and hard-to-find questions from open knowledge graphs. Second, we apply end-to-end multi-turn reinforcement learning (RL) to enhance LLMs' long-horizon reasoning with deep search. To encourage diversity and reduce redundancy, we design a redundancy penalty that discourages repeated similar queries. Experiments show that DeepDive-32B achieves a new open-source competitive result on BrowseComp, outperforming WebSailor, DeepSeek-R1-Browse, and Search-o1. We demonstrate that multi-turn RL training improves deep search ability and significantly contributes to the performance improvements across multiple benchmarks. We observe that DeepDive enables test-time scaling of tool calls and parallel sampling. Our code of DeepDive can be accessed at https://anonymous.4open.science/r/DeepDive-CAC6/.

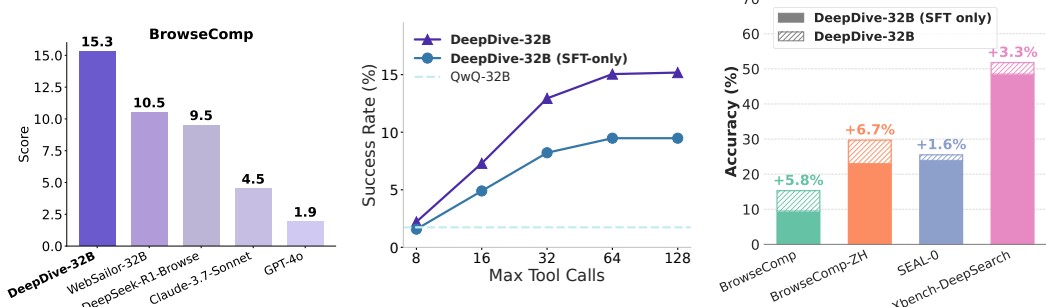

Figure 1: *Left*: DeepDive-32B outperforms open-source deep search and proprietary models on BrowseComp. *Middle*: DeepDive drives the model's deep search ability with maximum tool calls, which improves performance on BrowseComp. *Right*: Multi-turn reinforcement learning consistently enhances DeepDive-32B on four deep search benchmarks.

## 1 Introduction

Large language models (LLMs)—trained with reinforcement learning (RL) using verifiable rewards—have demonstrated strong performance in complex reasoning tasks, such as mathematics and coding competitions(Wei et al., 2022; DeepSeek-AI et al., 2025; OpenAI, 2024b; 2025; Grok, 2025). As real-world tasks become increasingly complex, integrating external tools like browsing expands an LLM's knowledge beyond its training corpus. This shift requires the LLM to execute as an autonomous *agent* capable of handing complex tasks.

Notably, deep search agents are expected to reason over and search from hundreds of online sources to locate complex, hard-to-find information, such as answering the questions in BrowseComp (Wei et al., 2025). However, open models fall far behind proprietary LLMs such as OpenAI DeepResearch

as deep search agents (Li et al., 2025b; Song et al., 2025; Li et al., 2025c; Wu et al., 2025a). We attribute this gap to the shortage of hard-to-find data and the absence of multi-turn RL training.

---

**BrowseComp**

Please identify the __fictional character__ who occasionally __breaks the fourth wall__ with the audience, has a backstory involving help from __selfless ascetics__, is known for his __humor__, and had a TV show that aired between the __1960s and 1980s__ with __fewer than 50 episodes__.

__Blurry Entity:__ require both Browse and Reason

Figure 2: An illustrative example of BrowseComp (Wei et al., 2025) questions, which often demand long-horizon reasoning and deep search integration across multiple blurry entities.

First, data-wise, most existing QA datasets usually feature relatively simple questions that do not reflect true "hard-to-find" cases. For example, questions in HotpotQA (Yang et al., 2018) can often be solved by searching for a few clear entities. In contrast, deep search questions such as those in BrowseComp usually involve multiple blurry entities, requiring long-horizon reasoning and deep search to reach the correct answer. Second, training-wise, how to effectively combine long-horizon reasoning with deep search tool use remains an open question. Even strong reasoning models such as DeepSeek-R1 (DeepSeek-AI et al., 2025) make only shallow tool calls and often suffer from hallucinations (see Figure 1 Left). In addition, existing browsing agents that integrate browsing tools are primarily designed to address direct search tasks. For example, systems like R1-Searcher (Song et al., 2025), ReSearch (Chen et al., 2025), and DeepResearcher (Zheng et al., 2025) are mainly trained and evaluated on datasets similar to HotpotQA, including 2WikiMultiHopQA (Ho et al., 2020), Bamboogle (Press et al., 2022), and Musique (Trivedi et al., 2022).

To address these challenges, we present DeepDive to advance deep search agents. First, we automatically generate challenging QA pairs from open knowledge graphs (KGs). Second, we use end-to-end multi-turn RL to improve long-horizon reasoning in deep search scenarios.

On the data side, we address the lack of difficulty in QA datasets by automatically constructing a deep search QA dataset from KGs, as they naturally support multi-hop connections, and each entity has different attributes. By deliberately blurring some attributes of each entity during question construction, we create a form of "blurry entity". We then perform random walks on the KG to extract long, multi-hop paths and use LLMs to further obfuscate key cues, making the QA pairs more challenging. This data synthesis process produces data that effectively stimulates LLMs' long-horizon reasoning and deep search abilities.

On the training side, we adopt end-to-end multi-turn RL training to integrate reasoning with search tool use. We employ the multi-turn GRPO (Shao et al., 2024) algorithm for RL, where the LLM interacts with a web environment and receives rewards based on the final answer in the constructed QA dataset. To encourage diverse exploration and prevent redundant search behavior, we design a redundancy penalty that discourages repeated similar queries as measured by Jaccard similarity. Figure 1 (middle) shows that the RL-trained model increases tool use more effectively than baselines during inference, demonstrating test-time scaling of tool calls for improved deep search.

The DeepDive method is trained on two open models: GLM-Z1-9B-0414 (GLM et al., 2024) and QwQ-32B (Team, 2025). The constructed data consists of 3,090 high-quality deep search QAs derived from KGs. Built on this, DeepDive-32B reaches an accuracy of 15.3% on BrowseComp, surpassing many open agents—WebSailor, Search-o1, and DeepSeek-R1-Browse—and achieving a new open-source competitive result (see Figure 1 left). Experiments demonstrate that the performance of the DeepDive models benefits significantly from the proposed end-to-end multi-turn RL training (see Figure 1 right). We further validate across multiple challenging deep search QA benchmarks and analyze test-time scaling for tool calls and parallel sampling.

The main contributions are summarized as follows:

- We propose an automated method to synthesize deep search QA pairs from open KGs.

- We introduce DeepDive, an end-to-end multi-turn RL framework with a redundancy penalty that encourages diverse, efficient search.

- Built on open models, DeepDive-32B achieves 15.3% on BrowseComp and shows strong test-time scaling in tool calls and parallel sampling.

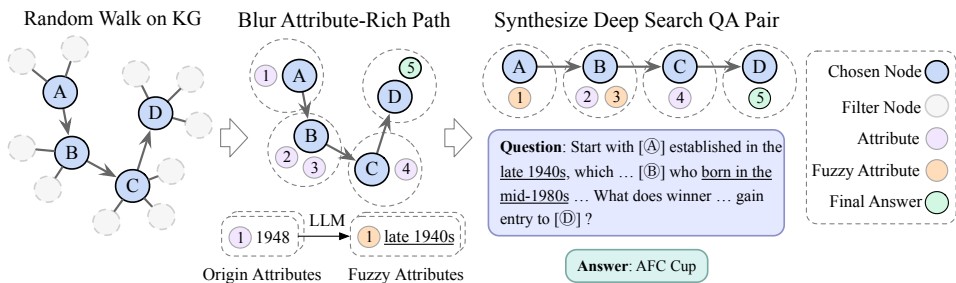

Figure 3: Overview of automated question–answer (QA) data synthesis from knowledge graphs (KGs) for DeepDive. Deep search QA pairs are automatically constructed by performing random walks over a knowledge graph and subsequently obfuscated using a large language model.

## 2 THE DEEPDIVE METHOD

We present DeepDive to advance the long-horizon information-seeking ability of deep search agents. In DeepDive, we introduce two techniques, targeting the data construction and RL stages, respectively. To generate a large-scale corpus of challenging deep-search QA pairs, we develop an automated and controllable data synthesis method with knowledge graphs (KG) (see Figure 3). To enhance the agent's capabilities for long-horizon reasoning and browsing, we leverage the constructed data to perform end-to-end multi-turn RL training (see Figure 4).

To mimic human web navigation, we establish an interaction framework as the learning environment for our deep search agent. The agent follows an iterative cycle of reasoning, tool execution, and observation, followed by ReAct (Yao et al., 2023) (see Figure 4).

Formally, at step $t$, the agent generates a chain-of-thought $c_t$, executes a browsing action $a_t$, and observes the web content $o_t$. This process repeats until the agent determines it has collected sufficient information and executes a terminating action $a_{\text{eos}}$ to give the final answer. The entire task execution can be represented as a trajectory $\mathcal{T}$:

$$\mathcal{T} = [q, (c_1, a_1, o_1), \ldots, (c_m, a_m, o_m), c_{\text{ans}}, a_{\text{eos}}], \quad m \leq n \tag{1}$$

The action $a_t$ is drawn from a browsing action space with three core operations: `search`, `click`, and `open`. A `search` action to retrieve web page summaries with given keywords, a `click` action to access specific pages from search results, and an `open` action to access specified URLs directly.

### 2.1 AUTOMATED DATA SYNTHESIS FROM KNOWLEDGE GRAPHS

Building deep search agents requires training data that goes beyond conventional multi-hop QA. While datasets like HotpotQA involve predictable reasoning steps, true deep search agents should act like human researchers who iteratively search, filter, and synthesize scattered evidence from the web. This thus calls for complex, difficult, and hard-to-find questions that even domain experts need hours to search and solve. Such complex training data is critical for developing agents to handle real-world tasks where information is scattered, conflicting, and hard to locate.

However, the specific training data required to cultivate this skill is naturally scarce on the internet. With manual annotation being prohibitively expensive and difficult to scale, synthetic data generation emerges as the most efficient and scalable solution.

**Knowledge Graphs with Hard-to-Find Information.** Naturally, knowledge graphs (KGs) provide a structured and semantically-rich environment for multi-hop reasoning, making them particularly well-suited for generating supervision data for training deep search agents. First, *verifiability*: KGs encode factual entity-relation triples that are inherently traceable and objective, ensuring answer correctness and significantly improving data reliability compared to fully model-generated QA pairs. Second, *multi-hop structure*: KGs allow us to explicitly control reasoning depth by performing random walks of varying lengths, enabling the generation of questions requiring multiple inference steps. Third, *reasoning controllability*: each entity node contains multiple attributes that can be

selectively obscured (such as dates, names, or locations), thereby increasing ambiguity and preventing models from exploiting shortcut solutions. This forces models to iteratively reason, search, and validate before finding answers. In light of these advantages, we propose an automated KG-based method to generate scalable, high-quality, and reasoning-intensive QA pairs.

**Automated Data Synthesis from KGs.** The main idea is to generate complex reasoning paths from KGs. A knowledge graph is a directed graph $G = (V, E)$ where $V$ represents entities and $E \subseteq V \times V$ represents relationships between them (Ji et al., 2021). Each entity $v_i \in V$ has associated attributes $A(v) = \left[a_i^0, a_i^1, \cdots, a_i^t\right]$.

To create questions that require deep reasoning and browsing, we generate paths by taking a random walk through the graph. Starting from an initial node $v_0$, we navigate through the graph for $k$ steps to form a path $P = [v_0, v_1, \ldots, v_k]$, where each step $(v_i, v_{i+1})$ is a valid edge in the graph. We choose a longer path length (e.g., $k > 5$) to increase the potential reasoning complexity. However, questions generated solely based on the node sequence $P$ tend to be too simple, similar to those in HotpotQA, as their answers can be found by direct search.

To further increase the complexity and ambiguity of the questions, we enrich and obfuscate the path by incorporating node attributes. Specifically, we combine each node $v_i$ in the path with its corresponding attributes to form an attribute-rich path $P_A$:

$$P_A = \left[\left(v_0, \left[a_0^0, a_0^1, \ldots\right]\right), \left(v_1, \left[a_1^0, a_1^1, \ldots\right]\right), \ldots, \left(v_k, \left[a_k^0, a_k^1, \ldots\right]\right)\right] \tag{2}$$

Subsequently, we select an attribute $a_k^i$ from the terminal node of the path, $v_k$, as the ground-truth answer. An LLM is then employed to obfuscate the information along the entire attribute-rich path $P_A$. This process involves techniques such as generalizing specific dates into ranges. The final output is a pair of challenging questions and answers $(q, a_k^i)$, generated as follows:

$$(q, a_k^i) = \text{LLM-obscure}(P_A) \tag{3}$$

**Improving Path Quality and Complexity.** In a graph random walk, each step directly impacts the quality of the final path, which in turn determines the complexity and logical soundness of the generated QA pair. To improve path quality, we apply two constraints to the random walk process.

First, we filter candidate nodes by setting an appropriate out-degree range $[d_{\min}, d_{\max}]$. If a node's out-degree is excessively high, it tends to be overly popular, making answers too predictable for the model. Conversely, nodes with low out-degree may hinder effective path expansion. Thus, the candidate set of nodes for the next step $\mathcal{N}(v_i)$ is defined as:

$$\mathcal{N}(v_i) = \{u \mid (v_i, u) \in E \land d_{\min} \leq d(u) \leq d_{\max}\} \tag{4}$$

Second, to ensure logical consistency of the path, we leverage an LLM to choose the next node. Given the current path $P_i = [v_0, \ldots, v_i], i < k$, the LLM evaluates all candidates in $\mathcal{N}(v_i)$ and selects the most relevant next node to the existing path as $v_{i+1}$:

$$v_{i+1} = \text{LLM-select}(P_i, \mathcal{N}(v_i)) \tag{5}$$

Together, these constraints guide the random walk to produce reasoning paths that are both complex and coherent, synthesizing high-quality QA pairs.

To further increase question difficulty, we implement an automated filter using a frontier model (e.g., GPT-4o (OpenAI, 2024a)) with basic search capabilities. Each question is tested four times—if the model solves it in any attempt, the question is discarded. Only questions that fail all four attempts are retained, ensuring our dataset contains exclusively challenging tasks requiring complex reasoning and advanced web browsing rather than simple information lookups.

## 2.2 END-TO-END MULTI-TURN REINFORCEMENT LEARNING

Given the challenging QA dataset, we use end-to-end multi-turn reinforcement learning (RL) to train the agent for deep search. Based on the standard GRPO algorithm for multi-turn RL, we enhance the reward mechanism by combining strict rewards for correctness with a redundancy penalty to encourage search diversity.

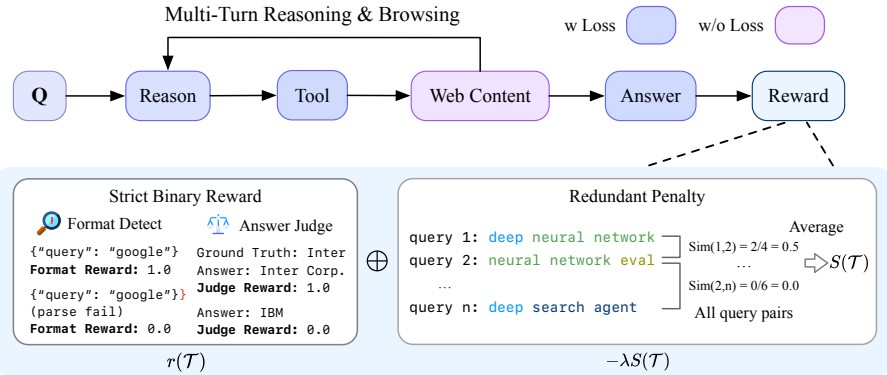

Figure 4: Overview of multi-turn RL in DeepDive.

**Multi-Turn RL.** Unlike single-turn RL, where the model outputs a single response per question, multi-turn RL lets the agent perform multiple reasoning and tool-use steps before arriving at a final answer. We employ the Group Relative Policy Optimization (GRPO) algorithm (Shao et al., 2024) to train the deep search agent. For each question $q$, we sample the tool calling trajectories $G$ from the current policy $\pi_\theta$. For each trajectory $\mathcal{T}$, we then calculate a normalized advantage $A_i = \left(r_i - \text{mean}\{r_k\}_{k=1}^G\right) / \text{std}\{r_k\}_{k=1}^G$, then the policy parameters $\theta$ are updated to maximize a clipped objective function with a KL penalty:

$$\mathcal{L}(\theta) = \frac{1}{G} \sum_{i=1}^{G} \left[\min\left(\rho_i A_i, \text{clip}\left(\rho_i, 1 - \epsilon, 1 + \epsilon\right) A_i\right) - \beta \text{KL}\left(\pi_\theta \| \pi_{\text{ref}}\right)\right] \quad (6)$$

This objective uses the importance ratio $\rho_i = \pi_\theta(\mathcal{T})/\pi_{\theta_{\text{old}}}(\mathcal{T})$, where $\epsilon$ controls the clipping range and $\beta$ weights the penalty for diverging from a reference policy $\pi_{\text{ref}}$.

**Encouraging Diverse Search with Redundancy Penalty.** Deep search tasks are inherently multi-turn, as formalized in Eq. 1. Deep search tasks benefit significantly from diverse exploration strategies, as different search queries can uncover complementary information and lead to a more comprehensive understanding. To promote such diversity, we design a reward mechanism that encourages browsing agents to explore varied search approaches while maintaining correctness.

Our approach combines two key components. First, we measure search diversity by analyzing how similar queries are within a search trajectory. Given a trajectory $\mathcal{T}$ with all search queries $Q = [q_1, q_2, \dots, q_T]$, where each query $q_i$ contains keywords $q_i = \{w_{i,1}, w_{i,2}, \dots, w_{i,n_i}\}$, we calculate the Jaccard similarity (Real & Vargas, 1996) between any two queries as: $\text{sim}(q_i, q_j) = |q_i \cap q_j|/|q_i \cup q_j|$. The overall similarity across all queries in the trajectory is then computed as:

$$S(\mathcal{T}) = \frac{1}{T(T-1)} \sum_{i \neq j} \text{sim}(q_i, q_j), \quad S(\mathcal{T}) \in [0, 1] \quad (7)$$

This metric equals 1 when all queries are identical and 0 when all queries are completely disjoint. Lower similarity indicates more diverse search exploration.

Second, we employ a strict binary reward to ensure trajectory correctness. A trajectory $\mathcal{T}$ receives a +1 reward only when every step is correctly formatted, including the reason $c_i$ and the action $a_i$, and the final answer $a_{\text{eos}}$ matches the ground-truth $a^*$. Since entities may have multiple valid representations, we use an LLM judge (Zheng et al., 2023) for answer verification. Formally, the binary reward is defined as:

$$r(\mathcal{T}) = \begin{cases} 1, & (\forall i, \text{Format}(c_i, a_i)) \wedge \text{Judge}(a_{\text{eos}}, a^*) \\ 0, & \text{otherwise} \end{cases} \quad (8)$$

We combine these components into our final reward function:

$$r'(\mathcal{T}) = r(\mathcal{T}) - \lambda \cdot S(\mathcal{T}) \quad (9)$$

where $\lambda < 1$ controls how much we reward diverse queries. This formulation encourages agents to explore a wider range of search strategies while maintaining a strong emphasis on the correctness of the final answer, thereby fostering more efficient and comprehensive search behaviors.

## 3 EXPERIMENTS

### 3.1 SETUP

**Benchmarks.** We evaluate DeepDive on four public and challenging deep search benchmarks: BrowseComp (Wei et al., 2025), BrowseComp-ZH (Zhou et al., 2025), Xbench-DeepSearch (Xbench-Team, 2025), and SEAL-0 (Pham et al., 2025).

**Data Synthesis Details.** We build synthetic datasets from two public knowledge graphs, KILT (Petroni et al., 2020) and AMiner(Tang et al., 2012). First, we generate long-chain paths via random walking with parameters set to $k \in [5, 9], d = 3, d_{min} = 4, d_{max} = 8$. We then use Gemini-2.5-Pro (Team et al., 2023), leveraging its superior long-context ability, to obscure entities and synthesize the QA pairs. This process yields 3,250 deep search QA pairs, which are randomly split into 1,016 samples for Supervised Fine-Tuning (SFT) and 2,234 for Reinforcement Learning (RL).

**Training Details.** We integrate the Serper API (Serper) for web search, which returns the top-10 pages for each query. The Jina API (Jina.ai, 2025) handles the click and open operations. Our training process follows recent RL approaches for large language models (Guo et al., 2025; Hou et al., 2025; Li et al., 2025a), starting with a cold-start phase. We leverage the Claude-4-Sonnet-Thinking model (Anthropic, 2025a), which has tool-calling capabilities, to interact with browsing tools and generate cold-start data through multiple attempts and reject sampling, yielding 858 high-quality SFT traces.

We choose two open models as our backbone models: GLM-Z1-9B-0414 (GLM et al., 2024) and QwQ-32B (Team, 2025). Each model is trained for 3 epochs with a global batch size of 32, a learning rate of $1 \times 10^{-5}$, and a maximum context length of 104,800.

During RL, we conduct training using the open-source Slime framework (Zhu et al., 2025) with all 2,234 data samples. The training configuration includes a rollout size of 8, 16 samples per prompt, a global batch size of 128, a temperature of 1.0, and a maximum context length of 50,000 tokens. We set the redundancy penalty coefficient to $\lambda = 0.1$. To promote exploration, we set the KL penalty coefficient to $\beta = 0$ (Vassoyan et al., 2025) and employ a learning rate of $1 \times 10^{-6}$.

**Evaluation.** For datasets and models with previously-reported scores, we directly adopt the results from their respective papers. For all other evaluations, we follow the LLM-as-Judge framework (Zheng et al., 2023), employing Llama-3.1-70B (Dubey et al., 2024) to assess whether a model's final output matches the ground truth answer. To speed up evaluation during reinforcement learning (RL) training, each checkpoint is assessed on a fixed, randomly pre-sampled subset of BrowseComp-266, with a maximum of 75 turns. Once training saturates, we evaluate later checkpoints on the full BrowseComp dataset (1,266 instances) with the turn limit raised to 128. For other benchmarks whose total size is below 300, we evaluate on the entire dataset. To reduce variance and improve robustness, every dataset is evaluated twice, and the average accuracy is reported as the final result.

### 3.2 OVERALL PERFORMANCE

Table 1 presents a comprehensive comparison between DeepDive and a range of baselines across four challenging deep search benchmarks. From the results, we draw the following key observations:

**Competitive among Open Deep Search Agents.** The DeepDive-32B model excels on four challenging deep search benchmarks. For the BrowseComp benchmark, it ranks just behind OpenAI's DeepResearch and far ahead of other open-source models or agents. While most open-source models score under 10% on BrowseComp, DeepDive-32B achieved 15.3%. It also shows clear advantages on SEAL-0 and XBench-DeepSearch, indicating effective use of browsing for complex reasoning. The results also highlight the power of reinforcement learning (RL). The 32B model with only SFT has already scored 9.5% on BrowseComp, RL then enhances the model's core ability to combine reasoning with search, resulting in stable performance growth over the SFT version on each benchmark. Notably, these gains are less pronounced for the smaller 9B model, potentially because of its limited reasoning capacity or a tendency to overfit on synthetic data during its training.

Table 1: Evaluation of deep search QA benchmarks. Accuracy(%) is reported. * represents reported performance from existing studies. **bold**: best among open-source models; underline: second best.

| Model | Reason | Browse | BrowseComp | BrowseComp-ZH | Xbench-DeepSearch | SEAL-0 |
|---|---|---|---|---|---|---|
| | | | *Proprietary Models* | | | |
| GPT-4o | ✗ | ✗ | 0.9* | 11.1 | 18.0* | 0.9 |
| GPT-4o | ✗ | ✓ | 1.9* | 12.8 | 30.0 | 9.1 |
| Claude-3.7-Sonnet | ✗ | ✗ | 2.3 | 11.8 | 12.0 | 2.7 |
| Claude-3.7-Sonnet | ✗ | ✓ | 4.5 | 14.2 | 29.0 | 14.4 |
| o1 | ✓ | ✗ | 9.9* | 29.1* | 38.0 | 11.7 |
| Claude-4-Sonnet-Thinking | ✓ | ✗ | 2.6 | 21.5 | 27.0 | 9.0 |
| Claude-4-Sonnet-Thinking | ✓ | ✓ | 14.7 | 30.8 | 53.0 | 37.8 |
| Grok-DeepResearch | ✓ | ✓ | - | 12.9* | 50.0* | - |
| DeepResearch | ✓ | ✓ | 51.5* | 42.9* | - | - |
| | | | *Open-Source Models* | | | |
| GLM-Z1-9B-0414 | ✗ | ✗ | 0.6 | 2.4 | 8.0 | 7.2 |
| GLM-Z1-9B-0414 | ✗ | ✓ | 0.6 | 1.7 | 3.0 | 2.7 |
| QwQ-32B | ✓ | ✗ | 1.7 | 13.5 | 10.7* | 5.4 |
| QwQ-32B | ✓ | ✓ | 1.3 | 14.5 | 27.0 | 4.5 |
| DeepSeek-R1-0528 | ✓ | ✗ | 3.2 | 28.7 | 37.0 | 5.4 |
| Search-o1-32B | ✓ | ✓ | 2.8* | 17.9* | 25.0* | - |
| WebDancer-32B | ✓ | ✓ | 3.8* | 18.0* | 39.0* | - |
| WebSailor-7B | ✓ | ✓ | 6.7* | 14.2* | 34.3* | - |
| WebSailor-32B | ✓ | ✓ | 10.5* | 25.5* | **53.3*** | - |
| DeepDive-9B (*sft-only*) | ✓ | ✓ | 5.6 | 15.7 | 35.0 | 15.2 |
| DeepDive-9B | ✓ | ✓ | 6.3 | 15.1 | 38.0 | 12.2 |
| DeepDive-32B (*sft-only*) | ✓ | ✓ | 9.5 | 23.0 | 48.5 | 23.9 |
| DeepDive-32B | ✓ | ✓ | **15.3** | **29.7** | 51.8 | **25.5** |

**RL Drives Deeper Search Strategies.** Figure 5 illustrates the effect of reinforcement learning on DeepDive-32B through two key metrics: model performance and tool call counts. Evaluation accuracy on a randomly sampled subset (BrowseComp-266) consistently improves, accompanied by rising tool usage, indicating that the model explores progressively deeper search strategies. These results demonstrate that reinforcement learning trained on our synthetic data successfully enhances both performance and search depth, with benefits generalizing to unseen samples.

Figure 5: Evaluation accuracy and tool calls during RL training on a random subset (BrowseComp-266).

### 3.3 TEST-TIME SCALING FOR DEEPDIVE

We evaluate the test-time scaling capabilities of our model from two perspectives: single attempt scaling by increasing the tool call budget, and multiple attempt scaling through parallel sampling with different answer selection strategies. These experiments demonstrate how additional computation at inference time can substantially improve model performance.

**Tool Call Scaling during Inference.** Figure 6a and 6b show the impact of increasing the maximum number of tool calls on BrowseComp and BrowseComp-ZH. Performance improves steadily as the tool call budget grows. When the tool call limit reaches 16 or more, DeepDive-32B trained with reinforcement learning clearly outperforms its SFT-only counterpart, demonstrating the benefit of RL

for tool call scaling. The dotted line indicates the QwQ-32B baseline, which is relatively low on both datasets. Although QwQ-32B achieves about 15 points on BrowseComp-ZH without tool use, our model surpasses this baseline once the tool call budget exceeds 16.

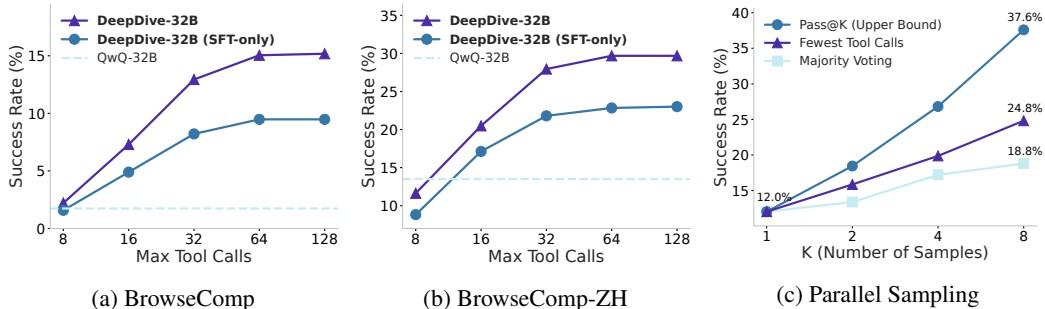

| (a) BrowseComp | (b) BrowseComp-ZH | (c) Parallel Sampling |

Figure 6: Test-time scaling results for DeepDive-32B. *Left* and *Middle*: Performance vs. maximum tool calls on BrowseComp and BrowseComp-ZH (x-axis in log scale). *Right*: Parallel sampling comparison on BrowseComp-266 (a randomly sampled subset), showing that selecting answers with the fewest tool calls outperforms majority voting.

**Parallel Sampling and Tool Call Voting.** Beyond scaling the number of tool calls, we investigate how parallel sampling can further improve performance. As shown in Figure 6c, majority voting, which selects the most frequent answer (Wang et al., 2022), improves DeepDive-32B performance on BrowseComp-266 from 12.0 to 18.8. We further analyze the distribution of tool calls across parallel samples and observe that answers requiring fewer tool calls before submission tend to be more accurate. This pattern likely occurs because the model stops earlier when confident in a good answer, whereas additional calls often reflect uncertainty and lead to less reliable results. Based on this observation, we propose selecting the answer with the fewest tool calls, which achieves a substantial improvement from 12.0 to 24.8, approaching the theoretical upper bound of 37.6 (pass@8).

## 3.4 ABLATION STUDY

**Reward Ablation** We evaluate two components of our reward design under identical RL settings: the strict format reward and the redundancy penalty. All models are assessed on BrowseComp-266 (a randomly sampled subset) at regular intervals (every 40 steps from 40 to 240). In Figure 7a, removing the format reward yields a curve that stays near 8.0 with almost no improvement, while adding the format reward produces a steady upward trend that remains about 2 absolute points higher throughout training. In Figure 7b and Figure 7c, adding the redundancy penalty increases accuracy in the later training phase (about 20%) and reduces tool call counts by roughly 14% under the same conditions. Overall, the strict format reward accelerates and stabilizes learning, and the redundancy penalty prunes redundant searches, improving search efficiency without sacrificing performance.

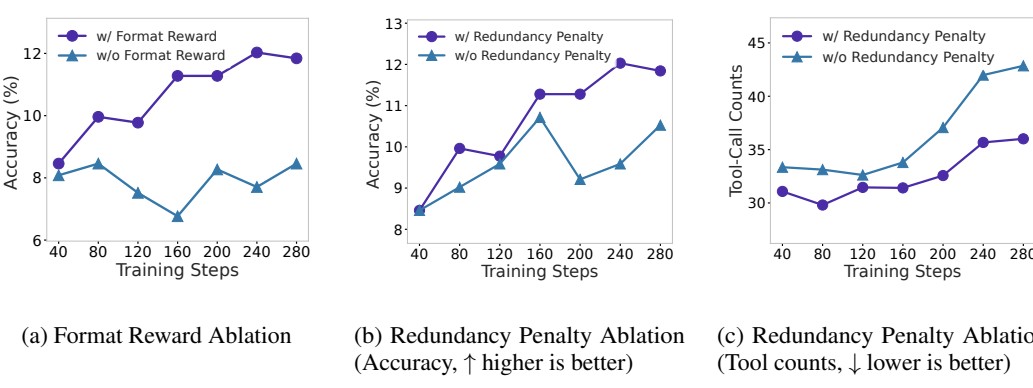

| (a) Format Reward Ablation | (b) Redundancy Penalty Ablation (Accuracy, ↑ higher is better) | (c) Redundancy Penalty Ablation (Tool counts, ↓ lower is better) |

Figure 7: Ablation of our reward design. All evaluations are on a sampled subset (BrowseComp-266).

**Synthetic Data Ablation**   We ablate our synthetic deep search QA data across SFT and RL, using the same four benchmarks as the main experiments. We report accuracy (Acc) and average tool calls (#Turn), where more tool calls indicate deeper search capability. As shown in Table 2, the base QwQ-32B performs poorly with low accuracy and almost no tool use. SFT with HotpotQA trajectories gives only modest gains, while SFT with synthetic data brings clear improvements across all benchmarks, boosting both accuracy and tool usage. For RL, fixing the best SFT model, HotpotQA yields only minor gains without changing usage patterns, whereas synthetic data drives large gains on both metrics, especially on BrowseComp-266. In summary, our synthetic data proves essential for both training stages, boosting performance and enabling long-horizon deep search capabilities.

Table 2: Ablation study of different training data. For efficiency, we evaluate a subset of BrowseComp (BrowseComp-266), while the other three benchmarks are evaluated in full.

| Backbone Model | Training Data | BrowseComp-266 | | BrowseComp-ZH | | XBench-DeepSearch | | SEAL-0 | |
|---|---|---|---|---|---|---|---|---|---|
| | | Acc | #Turn | Acc | #Turn | Acc | #Turn | Acc | #Turn |
| *Supervised Fine-tuning (SFT)* | | | | | | | | | |
| QwQ-32B | – | 1.9 | 1.5 | 14.5 | 1.2 | 27.0 | 1.5 | 4.5 | 1.1 |
| | + HotpotQA | 4.9 | 20.2 | 13.5 | 11.1 | 35.0 | 8.1 | 18.0 | 8.0 |
| | **+ our data** | **7.5** | **32.7** | **19.0** | **24.1** | **45.5** | **15.4** | **25.2** | **13.0** |
| *Reinforcement Learning (RL) from the best SFT model* | | | | | | | | | |
| DeepDive-32B (*SFT only*) | + HotpotQA | 9.2 | 33.2 | 22.7 | 23.3 | 47.0 | 15.1 | 21.6 | 13.6 |
| | **+ our data** | **12.0** | **47.2** | **29.7** | **24.9** | **50.0** | **16.7** | **25.5** | **14.5** |

## 4   RELATED WORK

**Reinforcement Learning for LLMs.**   Early reinforcement learning from human feedback (RLHF) demonstrated how human preferences could align models with user intent (Ouyang et al., 2022). Subsequent work shifted to verifiable reward signals to strengthen reasoning. Large-scale efforts such as OpenAI's o1 (OpenAI, 2024b) have empirically validated the effectiveness of verifiable-reward RL, while a wave of algorithmic improvements broadens the toolkit: GRPO (Shao et al., 2024) removes the critic model to simplify and stabilize training; DeepSeek's R1 (DeepSeek-AI et al., 2025) builds on GRPO to achieve strong reasoning performance; and DAPO (Yu et al., 2025) introduces fine-grained RL adjustments for scalable, robust pipelines.

**Deep Search Agents.**   ReAct Yao et al. (2023) first introduced a framework that combines reasoning and action steps, boosting LLM performance on complex tasks. Recent deep research agents, like DeepResearch (OpenAI, 2025) and Gemini Deep Research (Gemini, 2025), have reached near-expert levels in information seeking and reasoning. Proprietary systems like DeepResearch (OpenAI, 2025) and Gemini Deep Research (Gemini, 2025) reach near-expert levels. Open-source efforts include reinforcement learning approaches (ReSearch Chen et al. (2025), Search-o1 Li et al. (2025b), WebThinker Li et al. (2025c), DeepResearcher Zheng et al. (2025), Search-R1 Jin et al. (2025), and WebShaper Tao et al. (2025)) that optimize tool use and retrieval, and framework-based systems (OpenDeepResearch (Hugging Face, 2025), TTD-DR (Han et al., 2025)) that target long-form generation. A significant gap remains between open-source and proprietary models.

## 5   CONCLUSION

We present DeepDive that aligns deep reasoning with multi-turn web search through automated deep search QA synthesis and end-to-end multi-turn reinforcement learning. Our data pipeline generates ambiguity-rich, multi-hop questions with hidden cues, and our training introduces a redundancy penalty to encourage diverse and efficient search. After the RL stage, DeepDive-32B achieves 15.3% accuracy on BrowseComp, setting a new competitive standard for open-source models while surpassing larger agents and multiple strong proprietary baselines. Analyses show that complex supervision and multi-turn RL jointly ground tool use, that performance scales with tool-call budgets and parallel sampling, and that skills learned on hard problems transfer to simpler settings.

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

## A    BASELINES

We compare DeepDive against a diverse set of models, grouped into two categories:

- **Proprietary models:** This group includes both non-browsing and browsing-capable models. Non-browsing models consist of GPT-4o (OpenAI, 2024a), Claude-3.7-Sonnet (Anthropic, 2025b), Claude-4-Sonnet-Thinking (Anthropic, 2025a) and o1 (OpenAI, 2024b), which are evaluated solely on their internal reasoning abilities. Browsing-capable proprietary models include Grok-DeepResearch (x.ai, 2025), and OpenAI's Deep Research (OpenAI, 2025). Additionally, we extend select non-browsing models with our browsing tools to examine performance gains via standard function calls.

- **Open-source models:** This group includes recent high-performing open-source models, both with and without browsing capabilities. The non-browsing models consist of GLM-Z1-9B-0414 (GLM et al., 2024), DeepSeek-R1-0528 (Guo et al., 2025) and QwQ-32B (Team, 2025). We compare our method with recent open-source web agents, including Search-o1 (Li et al., 2025b), WebDancer (Wu et al., 2025a), and WebSailor (Li et al., 2025a). To ensure a fair comparison, we also enable standard function calling for GLM-Z1-9B-0414 and QwQ-32B, allowing them to browse during evaluation.

## B    GENERALIZATION ON SIMPLE SEARCH TASKS

While DeepDive is trained on synthetic data based on knowledge graphs for challenging tasks like BrowseComp and BrowseComp-ZH, we evaluate its performance on simpler search benchmarks: HotpotQA (Yang et al., 2018), Frames (Krishna et al., 2024), and WebWalker (Wu et al., 2025b), which involve more direct, less ambiguous questions. We compare DeepDive against two non-search models (o4-mini and DeepSeek-R1-0528) and two proprietary search-enabled models using the same search engine. We evaluate 512 randomly selected HotpotQA questions and full test sets for other benchmarks. Figure 8 shows that both DeepDive-32B (SFT-only)

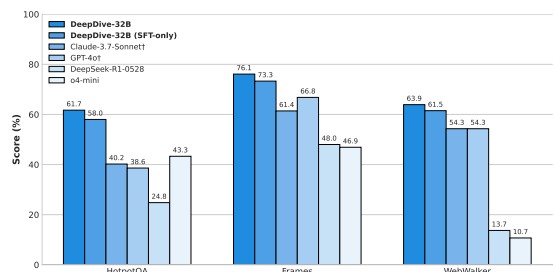

Figure 8: DeepDive-32B generalization on simple search benchmarks.

and DeepDive-32B outperform all baselines, with reinforcement learning providing additional improvements across all benchmarks. These results confirm DeepDive's strong generalization and search capabilities.

## C    ADDITIONAL STUDY: SEMI-AUTOMATED I.I.D. DEEP SEARCH QA SYNTHESIS FOR RL

We perform an additional study to directly improve model performance on deep search benchmarks. Straightforwardly, we can construct i.i.d. QA pairs with BrowseComp, whose questions are so challenging that expert annotators have to spend hours solving them, ensuring that simple search strategies are ineffective. However, reaching the depth and breadth of BrowseComp requires heavy human effort in research, annotation, and data curation. To reduce annotation costs, we present a semi-automated framework.

**i.i.d. Data Synthesis.**    We adopt a semi-automated framework to reduce the burden on annotators, where each annotator is supported by the OpenAI o3 model (OpenAI, 2025) equipped with search capabilities and follows a four-stage process.

First, based on the nine topical domains defined in BrowseComp, the annotator collaborates with the model to identify root domains that contain abundant factual and structured web content. Second, the annotator explores various linked pages within each root domain using the model's navigation

and search features, and selects verifiable entities along with their associated attributes. Third, the annotator conducts further targeted searches related to each selected entity and engages in multi-turn interactions with the model to construct new challenging multi-hop questions. These questions are carefully written to obscure key information while retaining verifiability. Fourth, the annotator uses the model to attempt to answer the synthetic question. If the answer is incorrect or if multiple plausible answers exist, the sample is discarded. The annotator also records the time taken by the model to arrive at an answer in order to identify questions that are more difficult and of higher quality. This workflow requires minimal prior knowledge from the annotator.

Through iterative model-guided discovery, question construction, and verification, annotators can efficiently produce complex, high-quality deep search QA pairs. The same procedure is applied to Chinese websites to enhance multilingual performance. As a result, we obtain a total of 2,997 English and 275 Chinese challenging deep search QA pairs.

**RL with i.i.d. Deep Search Data.** We follow the same pipeline as Section 3, using SFT for cold-start and difficulty-based filtering to build a high-quality subset for RL, with all training configurations remaining identical to those before, except for the data. Table 3 presents the performance after incorporating i.i.d. training data. Notably, the DeepDive-32B-RL model achieves an accuracy of 20.8% on the full BrowseComp benchmark, representing a 40% improvement over the previous best score of 15.3% and significantly outperforming open-source alternatives. Owing to the inclusion of Chinese content in the new training corpus, the new model also demonstrates considerable gains on Chinese-language benchmarks, namely BrowseComp-ZH and Xbench-DeepSearch. Interestingly, performance on SEAL-0 remains largely unchanged, which we attribute to the dataset's focus on recognizing and selecting among different search results, which is a challenge that highlights a key area for future model enhancement.

Table 3: Effect of i.i.d. deep search QA data for DeepDive. DeepDive-32B Accuracy (%) on 4 deep search benchmarks with and without i.i.d. data. **bold**: best performance; underline: second best.

| Model | data | BrowseComp | BrowseComp-ZH | Xbench-DeepSearch | SEAL-0 |
|---|---|---|---|---|---|
| DeepDive-32B (*sft-only*) | KG data | 9.5 | 23.0 | 48.5 | 23.9 |
| DeepDive-32B | KG data | 15.3 | 29.7 | 50.0 | **25.5** |
| DeepDive-32B (*sft-only*) | i.i.d data | 11.4 | 26.6 | 47.5 | 22.5 |
| DeepDive-32B | i.i.d data | **22.2** | **33.9** | **56.0** | 23.0 |

## D  DATA CONTAMINATION ANALYSIS

To ensure that the performance improvements are not the result of data leakage, we follow the contamination analysis protocol introduced in LLaMA 2 (Touvron et al., 2023) and evaluate the Human-in-the-Loop dataset used for training. For each evaluation sample, we tokenize the input (excluding special tokens) and extract all contiguous 10-token n-grams. A token is considered contaminated if it appears in any n-gram also found in the training corpus. The contamination rate for a sample is defined as the proportion of contaminated tokens. Based on these rates, we categorize each sample into four non-exclusive subsets: Clean (less than 20% contamination), Not Clean (20% or more), Not Dirty (less than 80%), and Dirty (80% or more). As shown in Table 4, more than 97% of the samples in the dataset are classified as Clean, and there are no samples in the Dirty category. The results indicate that there is almost no test-data leakage in the constructed dataset for training.

Table 4: Contamination analysis of BrowseComp evaluation samples using different synthetic data. Each sample is categorized based on the proportion of overlapping n-grams with the training set.

| Data Type | Contamination Rate | Clean | Not Clean | Not Dirty | Dirty |
|---|---|---|---|---|---|
| KG | 2.6 | 99.0 | 1.0 | 100.0 | 0.0 |
| i.i.d. | 3.4 | 97.7 | 2.3 | 100.0 | 0.0 |

# E    CASE STUDY

**Reinforcement Learning Reshapes the Model's Search Strategy**    Based on the sustained performance improvement on the BrowseComp-266 evaluation set during RL training, we study the model's search behavior because most of its actions involve issuing retrieval queries. Similar to human interaction with contemporary search engines like Google Search, which allow exact match quoting, logical OR aggregation and term exclusion with a leading minus sign, the retrieval interface used during training and evaluation supports these same advanced features. We therefore collected every query generated by the model when solving the evaluation set and calculated three metrics: (1) *Quote Usage*: the fraction of queries containing double quotes for exact phrase matching; (2) *Minus Usage*: the fraction of queries containing a leading minus sign to exclude terms. (3) *OR Usage*: the fraction of queries containing the OR operator to combine alternative terms; The evolution of these metrics over training steps is plotted in Figure 9.

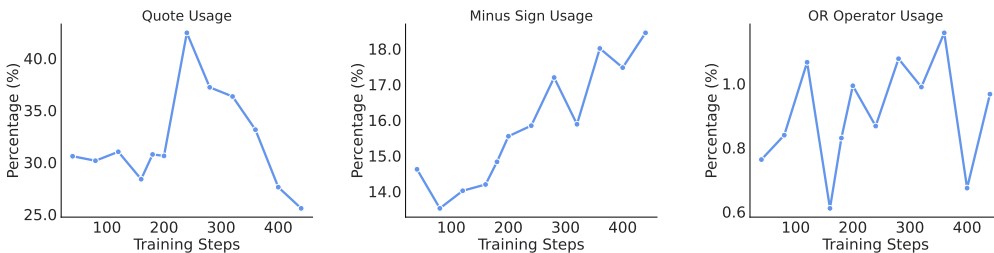

Figure 9: Evolution of *Quote Usage*, *Minus Usage* and *OR Usage* over RL training steps on BrowseComp-266.

From Figure 9, we observe that Quote Usage increases from around 30% to 40% at the early stage of training, then gradually decreases to below 25%. OR Usage steadily grows from approximately 2% to 8%. In contrast, Minus Usage continues to rise from 14% to 18% throughout the training. This trend suggests that the model initially learns to adopt the quoting strategy early in reinforcement learning, but its advantage becomes less prominent over time, leading to a decline in usage. Meanwhile, the model steadily improves its ability to use minus operators, and OR Usage remains stable between 0.8% and 1%, indicating limited but consistent application.

# F    USE OF LLMS

Large language models (LLMs) were used solely for language polishing and grammar refinement during manuscript preparation. All research ideas, methodologies, experiments, and analyses were independently conceived, designed, and validated by the authors.

