# OpenReview forum: "DeepDive: Advancing Deep Search Agents with Knowledge Graphs and Multi-Turn RL"
_ICLR.cc/2026/Conference — ICLR 2026 Conference Withdrawn Submission_

### Official Review · Reviewer_zmhg · 2025-10-30

**Soundness:** 3
**Presentation:** 2
**Contribution:** 2
**Rating:** 2
**Confidence:** 4

**Summary:**

This paper presents DeepDive, a method to enhance the capabilities of LLMs on complex "deep search" tasks, where they are often limited by long-horizon reasoning and a lack of quality training data. The approach has two key components:
1) A strategy to automatically synthesize complex, obfuscated question-answer pairs from Knowledge Graphs by using random walks and LLM-based entity blurring; and 2) An end-to-end multi-turn RL framework based on GRPO, which introduces a "redundancy penalty" to encourage diverse search.

Experiments show that DeepDive-32B achieves competitive results among open-source models on the BrowseComp benchmark.

**Strengths:**

- The paper's primary contribution is an automated data synthesis method for deep search. By leveraging the multi-hop structure and attributes of Knowledge Graphs (KGs) and using an LLM for "obfuscation," this method addresses the critical lack of "hard-to-find" questions in existing datasets.

- The paper provides detailed analyses, including scalability with tool calls, parallel sampling strategies, and the evolution of the model's search behavior during RL training.

**Weaknesses:**

- A significant concern arises from Appendix C, which introduces a semi-automated, i.i.d. data synthesis strategy. This method, using only ~3,000 QA pairs, achieves far superior results (22.2% on BrowseComp) compared to the main paper's automated KG method (15.3%).  This large performance discrepancy calls into question the true value and scalability of the main proposed KG method.

- The novelty of the RL framework is limited. The core training framework is based on the existing GRPO algorithm, with the primary addition being a "Redundancy Penalty." While effective, this appears to be more of an effective engineering trick rather than a substantial algorithmic contribution.

- The ablation study for the "Format Reward" (Figure 7a) is inconclusive. The results show that removing it causes learning to stagnate (around 8.0% accuracy). This only demonstrates its necessity within the current framework, not its superiority or design effectiveness. The authors fail to compare it against other alternative intermediate rewards.

**Questions:**

1. Regarding the large performance gap between the main KG method (15.3%) and the appendix i.i.d. method (22.2%): Since 3k QA pairs is not that much, can the authors elaborate on the cost-benefit trade-off? Specifically, what was the human-hour cost for the ~3k i.i.d. samples compared to the compute cost (including LLM obfuscation and filtering) for the KG data?

2. Regarding the Format Reward: As the ablation shows, this reward is critical for learning. Can you compare it against simpler baselines (e.g., a sparse reward given only for a successfully parsed tool call, rather than the strict, full format check)? This would help clarify if the strictness of the reward is the key factor, or just the presence of an intermediate signal.

---

> ### Author Response · Authors · 2025-11-21
> **Response to Reviewer zmhg (1/2)**
>
> Thank you for the constructive review. We now offer point-by-point replies to elaborate further and clarify our contributions.
>
> > **W1: A significant concern arises from Appendix C, which introduces a semi-automated, i.i.d. data synthesis strategy. This method, using only ~3,000 QA pairs, achieves far superior results (22.2% on BrowseComp) compared to the main paper's automated KG method (15.3%). This large performance discrepancy calls into question the true value and scalability of the main proposed KG method.**
> >
> > **Q1: Since 3k QA pairs is not that much, can the authors elaborate on the cost-benefit trade-off? Specifically, what was the human-hour cost for the ~3k i.i.d. samples compared to the compute cost (including LLM obfuscation and filtering) for the KG data?**
>
> For the ~3k QA pairs, the two pipelines have very different cost–benefit profiles in terms of **human-cost** and **scalability**:
>
> | **Type** | **Construction**                           | **Human Cost**                     | **Scalability**                |
> | -------- | ------------------------------------------ | ---------------------------------- | ------------------------------ |
> | KG       | Automatic walk + LLM obfuscation/filtering | ~0 human hours                     | **1000** samples/day           |
> | i.i.d.   | Experts + LLM on real web pages            | 5 annotators, several weeks for 3k | **100** samples/week/annotator |
>
> **For KG data (main method):** The KG pipeline performs controlled random walks on a structured KG (CPU-only), then applies LLM-based filtering and obfuscation. Once the pipeline is implemented, generating ~3k KG QA pairs is a matter of running the pipeline for a few days, with no additional human annotation.
>
> **For i.i.d. data (appendix method):** The i.i.d. data is semi-automated and requires more human intervention. To ensure both difficulty and answerability for deep search, skilled annotators should navigate real web pages, select entities, design questions with the help of a strong model (e.g., o3), and manually verify answers. Producing the same ~3k examples requires several person-weeks of expert effort.
>
> In this sense, the **i.i.d. data is a small, expensive source**, whereas the KG data is a large, **fully automated source**. It is natural that, at the same small scale (~3k), the curated i.i.d. data achieves higher performance (22.2% vs. 15.3% on BrowseComp). However, the KG pipeline can be applied to new knowledge graphs and **run at a much larger scale without extra labeling effort**, making it a promising route for building a large, agentic, continual-pretraining-level corpus.
>
> > **W2: The novelty of the RL framework is limited. The core training framework is based on the existing GRPO algorithm, with the primary addition being a "Redundancy Penalty." While effective, this appears to be more of an effective engineering trick rather than a substantial algorithmic contribution.**
>
> We would like to clarify that there may be a misunderstanding. The novelty of our RL framework lies in mitigating the **oversearch** phenomenon that arises when training deep-search agents using RL. We then introduce a redundancy penalty to address this **efficiency problem**.
>
> Specifically, on benchmarks such as BrowseComp, we found that as RL training continues, the average number of tool calls per question rises from approximately 30 to 42. Many of these extra queries are **near-duplicates** with almost identical web content, wasting API costs and long inference times without adding useful information. **Similar overthinking effects** have been observed in reasoning models, such as DeepSeek-R1 [1], yet no prior work offers a solution in the deep search setting.
>
> We therefore added a redundancy penalty to the whole trajectory reward, discouraging trajectories with many similar queries. **The average number of tool calls** on BrowseComp falls to 36, **a reduction of roughly 14%**, which equals about 8k fewer calls (including search APIs and long-context inference) across the full evaluation:
>
> | **Setting**         | Accuracy  $\uparrow$ | **Redundancy penalty** | **Avg. Tool Calls**  $\downarrow$ |
> | ------------------- | -------------------- | ---------------------- | --------------------------------- |
> | GRPO (vanilla)      | 14.8                 | ✗                      | 42                                |
> | **DeepDive (ours)** | **15.3**             | ✓                      | **36**                            |
>
> In addition, at the data construction stage, we design a **controllable** data construction pipeline that **explicitly requires deeper search** by constructing longer paths with multiple obfuscated entities. As a result, the agent naturally invokes more tool calls to solve each question. This more complex data both improves performance (e.g., BrowseComp from 9.5 to 15.3) and increases tool usage, which is **directly connected to the over-search phenomenon we observe**.

---

> > ### Author Response · Authors · 2025-11-21
> > **Response to Reviewer zmhg (2/2)**
> >
> > > **W3: The ablation study for the "Format Reward" (Figure 7a) is inconclusive. The results show that removing it causes learning to stagnate (around 8.0% accuracy). This only demonstrates its necessity within the current framework, not its superiority or design effectiveness. The authors fail to compare it against other alternative intermediate rewards.**
> > >
> > > **Q2: Regarding the Format Reward: As the ablation shows, this reward is critical for learning. Can you compare it against simpler baselines (e.g., a sparse reward given only for a successfully parsed tool call, rather than the strict, full format check)? This would help clarify if the strictness of the reward is the key factor, or just the presence of an intermediate signal.**
> >
> > Following your suggestion, we implement a simpler sparse-format baseline and compare it directly with our strict design under the same RL framework setup (i.e., the same model, data, and hyperparameters).
> >
> > In the **strict** version, any trajectory that contains a format or parsing error (e.g., an invalid tool call) is terminated early and assigned zero reward for the entire trajectory. In the **sparse** baseline, trajectories with format errors are dropped from the batch and replaced by resampled, format-correct rollouts; they receive no explicit penalty, only the absence of reward. The **performance comparison at 32B scale:**
> >
> > | **Reward Design** | **BrowseComp** | **BrowseComp-ZH** | **XBench-DeepSearch** | **SEAL-0** |
> > | ----------------- | -------------- | ----------------- | --------------------- | ---------- |
> > | **Full (strict)** | **15.3**       | **29.7**          | 51.8                  | **25.5**   |
> > | Sparse            | 13.1           | 24.8              | **52.5**              | 22.3       |
> >
> > We also measure the proportion of trajectories with format errors on BrowseComp:
> >
> > | **Reward Design** | **Format Rrror Ratio** $\downarrow$ |
> > | ----------------- | ----------------------------------- |
> > | **Full (strict)** | **18.7%**                           |
> > | Sparse            | 29.9%                               |
> >
> > Across three of the four benchmarks, **the strict reward achieves higher accuracy** (for example, +2.2 on BrowseComp and +4.9 on BrowseComp-ZH). It also **reduces the format error rate by more than 10 percent**. The strict format reward consistently outperforms the sparse baseline, indicating that the gain stems from strictly enforcing correct tool calls rather than merely providing an intermediate reward signal.
> >
> >
> > **Reference**
> >
> > [1] Sui et al. Stop Overthinking: A Survey on Efficient Reasoning for Large Language Models. arXiv preprint arXiv:2503.16419, 2025.

---

### Official Review · Reviewer_6txM · 2025-10-31

**Soundness:** 2
**Presentation:** 3
**Contribution:** 2
**Rating:** 4
**Confidence:** 4

**Summary:**

This paper introduces DeepDive, a framework designed to improve the "deep search" capabilities of open-source Large Language Models (LLMs), turning them into more effective web-browsing agents for solving complex problems.

Key Contributions：
* Automatic Data Synthesis: It proposes a novel strategy to automatically generate complex, difficult, and hard-to-find question-answer pairs from open knowledge graphs. This creates a rich dataset for training without manual effort.

* Reinforcement Learning (RL) for Training: It employs an end-to-end multi-turn Reinforcement Learning process to train the LLM. This method enhances the model's long-horizon reasoning. It also introduces a "redundancy penalty" to discourage the agent from making repetitive or similar search queries, encouraging more efficient and diverse exploration.

**Strengths:**

* The knowledge graph-based QA synthesis ensures high-quality, reasoning-intensive training data—by blurring entity attributes and filtering via frontier models (e.g., GPT-4o), it creates "hard-to-find" questions that truly stimulate deep search capabilities.

* The multi-turn RL framework (with GRPO algorithm and redundancy penalty) encourages diverse, efficient search: the penalty discourages repeated queries (measured by Jaccard similarity), while the end-to-end design integrates reasoning and tool use, enhancing long-horizon search.

* Strong experimental validation: DeepDive-32B sets a competitive open-source record on BrowseComp (15.3% accuracy), outperforms other open agents, and demonstrates test-time scalability (better performance with more tool calls/parallel sampling).

**Weaknesses:**

Many of the contributions in this paper lack originality and novelty. For specifics, please refer to the "Questions" section below.

**Questions:**

Many of the paper's contributions seem to lack novelty. For example:

* One of the core claimed contributions is the synthesis of QA pairs from KGs with an obfuscation step. However, this approach was already proposed in the prior work, WebSailor. What is the specific contribution of this paper in this regard?
* There is extensive prior work on applying reinforcement learning (RL) to search agents. What is the novelty and contribution of the RL framework presented here compared to existing methods?

---

> ### Author Response · Authors · 2025-11-21
> **Response to Reviewer 6txM (1/2)**
>
> We are grateful for your constructive suggestions and respond to them point by point below to further clarify our contributions.
>
> > **One of the core claimed contributions is the synthesis of QA pairs from KGs with an obfuscation step. However, this approach was already proposed in the prior work, WebSailor. What is the specific contribution of this paper in this regard?- There is extensive prior work on applying reinforcement learning (RL) to search agents.**
>
> DeepDive utilizes controlled KG random walks with entity obfuscation to **synthesize more challenging deep-search QA data**. Compared to WebSailor, we explicitly increase both the path depth and the number of **fuzzy entities** per question **(5.2 vs. 3.2)**. Under the same fair evaluation, our data is **more difficult and requires deeper search**. There are two main methodological differences from WebSailor.
>
> **First, the path construction differs.** WebSailor samples neighbors probabilistically on a Wikipedia KG. DeepDive instead builds longer chains by filtering out high-degree hub nodes and very low-degree dead ends, which tend to create shortcuts or trivial paths. Among the remaining neighbors, an LLM is used to select the most semantically relevant next entity. This produces longer and more coherent reasoning paths that better match real deep-search behavior.
>
> **Second, the obfuscating strategy differs**. Along these deeper chains, DeepDive hides multiple entities in each question by fuzzing attributes such as exact years or precise locations, while keeping the KG relations and final answers unchanged. This directly increases the number of fuzzy entities per question, making it much harder to answer by simple surface search.
>
> We also provide a **quantitative comparison**. Using all 20 released WebSailor examples and 32 randomly sampled DeepDive examples from our full set of 3,250, we ask QwQ-32B to count fuzzy entities per question and use DeepSeek-R1-0528 (direct reasoning mode) to measure the proportion of questions solved (averaged over 4 runs):
>
> | **Method**      | **Fuzzy Entity Number** $\uparrow$ | **R1 Accuracy** $\downarrow$ | **Open Data Size** |
> | --------------- | ---------------------------------- | ---------------------------- | ------------------ |
> | WebSailor       | 3.2                                | 40.0                         | 20                 |
> | DeepDive (ours) | **5.2**                            | **32.8**                     | **3250**           |
>
> Compared to WebSailor, DeepDive has **more fuzzy entities per question (5.2 vs. 3.2) and a lower R1 accuracy (32.8% vs. 40.0%)**, indicating that the questions are **harder** for R1 to answer through reasoning. Additionally, it releases all 3k QA pairs. Overall, this makes DeepDive **a larger, more ambiguous, and more challenging** deep-search QA dataset that is better suited for training long-horizon RL search agents.

---

> > ### Author Response · Authors · 2025-11-21
> > **Response to Reviewer 6txM (2/2)**
> >
> > > **Q2: What is the novelty and contribution of the RL framework presented here compared to existing methods?**
> >
> > Our RL framework targets **long-horizon deep search** and **explicitly optimizes format correctness and search efficiency**, which is not done in prior systems such as search-o1 or WebSailor.
> >
> > | **Method** | **Deep Search** | **Synthetic Data** | **Strict Format Reward** | **Search Efficiency**  |
> > | ---------- | --------------- | ------------------ | ------------------------ | ---------------------- |
> > | Search-o1  | ✗               | ✗                  | ✗                        | ✗                      |
> > | WebSailor  | ✓               | ✓                  | ✗ (soft format reward)   | ✗                      |
> > | **DeepDive (ours)**   | ✓               | ✓                  | ✓ (binary, strict)       | ✓ (redundancy penalty) |
> >
> > **For format reward**, any trajectory with format errors (in reasoning or function-call schema) is terminated and assigned **zero reward**. This binary rule makes format correctness a hard constraint, rather than a soft format term, as in WebSailor, which in practice both **stabilizes RL** (resulting in **monotonic learning curves**) and **speeds up training by cutting off invalid trajectories early**.
> >
> > **For search efficiency**, in deep-search tasks like BrowseComp, we observe that as the RL agent progresses, it tends to issue many **near-duplicate queries** (e.g., reordering keywords or changing one or two words) that lead to highly overlapping web pages. We address this by introducing a **trajectory-level redundancy penalty** that penalizes repeated or highly similar queries, thereby subtracting it from the final reward and explicitly encoding search efficiency into the RL objective.
> >
> > Starting from DeepDive SFT (9.5 on BrowseComp), correctness-based RL raises performance to **14.8** but uses about **42** tool calls per question. Adding the redundancy penalty keeps or slightly improves accuracy while **reducing tool calls from 42 to 36** (about 14% fewer calls).

---

> > > ### Comment · Reviewer_6txM · 2025-11-25
> > > **Official Comment by Reviewer 6txM**
> > >
> > > Thank you for the rebuttal.
> > >
> > > 1. It appears that one of your primary claimed contributions—synthesizing deep search QA pairs from open KGs—represents only an incremental advancement over WebSailor. The novelty seems limited to modifications in the sampling methods and fuzzing strategies.
> > >
> > > 2. Furthermore, the "strict format score" has already been proposed in prior work (e.g., Webdancer). The authors' contribution within the RL framework seems primarily confined to adding a "trajectory-level redundancy penalty" during the reward shaping phase.

---

> > > > ### Author Response · Authors · 2025-11-26
> > > > **Response to Reviewer 6txM**
> > > >
> > > > Thank you for raising these concerns about novelty.
> > > >
> > > > A fundamental difference between DeepDive and WebSailor lies in the KG structure used for data construction and the redundancy-aware RL objective to address the challenges of searching repeated or similar terms.
> > > >
> > > > On the data side, WebSailor performs a random walk on a local subgraph, while DeepDive walks along a single long chain (5–9 hops). In other words, WebSailor mainly explores a small local region of the KG, whereas DeepDive deliberately walks deeper and then fuzzes each node using its attributes. This produces questions with more fuzzy entities (5.2 vs 3.2) and greater difficulty (R1 solve rate 32.8 vs 40.0), which in turn encourages the search agent to perform deeper search.
> > > >
> > > > On the RL side, DeepDive introduces:
> > > >
> > > > - A redundancy-aware objective. We observe an over-search phenomenon in deep search, where the agent continues to search for near-duplicate queries that return almost the same web pages. We add a trajectory-level redundancy penalty that discourages such queries, reducing search-tool calls by about 14% while keeping accuracy.
> > > > - A strict format reward for stable training. In DeepDive, if any tool call in a trajectory has a format error, we immediately stop that trajectory and assign zero reward. This differs from WebSailor and WebDancer, which treat the binary format reward as a small term (e.g., $R=0.1\cdot r_{\text{format}}+0.9\cdot r_{\text{judge}}$), so trajectories with format errors can still get high rewards. Our stricter rule truncates invalid rollouts early, making training more stable.

---

### Official Review · Reviewer_GkUg · 2025-11-01

**Soundness:** 3
**Presentation:** 3
**Contribution:** 3
**Rating:** 6
**Confidence:** 3

**Summary:**

The paper proposes DeepDive, a web-browsing deep search agent. DeepDive develops a method to generate synthesized, hard-to-find QA data derived from knowledge graphs via long random walks with attribute obfuscation, and also combines a multi-turn RL using GRPO plus a redundancy penalty. The method aims to improve long-horizon reasoning with tools in a ReAct style agent. DeepDive-32B shows superior performance comparing to open models, though it still exists significant gaps comparing to GPT deep research.

**Strengths:**

1. The paper combines KG based synthesis of harder, multi-hop queries with multiturn RL, and demonstrates meaningful improvements on challenging deep web-search tasks, which shows that tougher supervision plus RL can better exploit tools and sustain longer reasoning chains.
2. The ablation studies isolate the effects of the reward design and the synthetic data generation, providing evidence that each component contributes measurably to performance and search efficiency.

**Weaknesses:**

1. Uncontrolled inference-time scaling. The paper does not standardize or report consistent test-time budgets across baselines (e.g., max tool-call limits). Several baseline scores appear to be taken from prior work rather than re-evaluated under a matched inference budget, making the headline comparisons confounded by inference-time scaling.
2. Incremental contribution in data generation. The proposed synthetic data pipeline generates multi-hop questions/targets by graph traversal (random walks) to generate search paths and then obfuscating attributes. Many existing efforts already use KG/graph path extraction to script multi-step reasoning trajectories and there are also works using LLM to paraphrase and harden queries.
3. Use recent models. The experiments may also evaluate on the Qwen3-32B model, which is enhanced on tool use ability comparing the QwQ-32B

**Questions:**

The authors can address the concerns mentioned in the above weaknesses 1 and 3.

---

> ### Author Response · Authors · 2025-11-21
> **Response to Reviewer GkUg (1/2)**
>
> Thank you for your helpful and constructive comments. We address them point by point below to provide a clearer explanation of our contributions.
>
> > **W1 & Q1: Uncontrolled inference-time scaling. The paper does not standardize or report consistent test-time budgets across baselines (e.g., max tool-call limits). Several baseline scores appear to be taken from prior work rather than re-evaluated under a matched inference budget, making the headline comparisons confounded by inference-time scaling.**
>
> We agree that for multi-step web agents, the maximum tool counts and context length are critical, and we explicitly study their effects in Section 3.3. Below, we report our main comparison to open-source methods under a matched tool-call budget:
>
> | **Model**           | **BrowseComp**  | **Max Tool Counts** | **Context Length** |
> | ------------------- | --------------- | ------------------- | ------------------ |
> | Search-o1-32B       | 2.8             | 10                  | 32k                |
> | WebDancer-32B       | 3.8             | -                   | 128k               |
> | WebSailor-32B       | 10.5            | ~32                 | 32k                |
> | DeepDive-32B (ours) | **12.9 (+23%)** | 32                  | 128k               |
> | DeepDive-32B (ours) | 15.3            | 128                 | 128k               |
>
> **For WebSailor-32B**, the paper specifies a 32k context window but does not explicitly mention the tool counts. Since one tool call plus its returned web content is roughly 1k tokens, this corresponds to an effective budget of approximately 32 tool calls.
>
> **For other open-source models**, which have significantly weaker deep-search capabilities, we observe that they typically terminate after only 3–5 tool calls and all achieve BrowseComp scores below 3.5; therefore, we exclude them from the comparison.
>
> Under a matched budget of **32 tool calls, DeepDive-32B still outperforms the strongest open-source baseline**, WebSailor-32B, on BrowseComp **(12.9 vs. 10.5)**. When we increase DeepDive’s budget to 128 tool calls, the score further improves to 15.3, suggesting that the model can effectively leverage additional search capacity. In the revised version, we will make these budget choices explicit in the main text and provide the full comparison table in the appendix.
>
>
>
> > **W2: Incremental contribution in data generation. The proposed synthetic data pipeline generates multi-hop questions/targets by graph traversal (random walks) to generate search paths and then obfuscating attributes. Many existing efforts already use KG/graph path extraction to script multi-step reasoning trajectories and there are also works using LLM to paraphrase and harden queries.**
>
> Compared to the strong KG-based data generation baseline WebSailor, our contribution to data generation is to **make the synthetic questions more challenging for deep search**. We use controlled KG random walks with entity obfuscation to construct QA pairs, and instead of sampling arbitrary multi-hop paths, we (i) **constrain the random walk** by out-degree and let an LLM choose the next entity to keep the path coherent, and (ii) restrict generation to longer reasoning chains (**more than 5 hops**) while obfuscating multiple entities along each path, so that each question contains several uncertain entities that must be resolved via deeper search.
>
> To compare our data with WebSailor, we use all 20 released WebSailor examples and 32 randomly sampled DeepDive examples (from our full set of 3,250). We prompt QwQ-32B to count the number of fuzzy entities per question and measure the proportion of questions solved by DeepSeek-R1-0528 (averaged over 4 runs):
>
> | **Method**      | **Fuzzy Entity Number**  $\uparrow$ | **R1 Accuracy** $\downarrow$ | **Open Data Size** |
> | --------------- | ----------------------------------- | ---------------------------- | ------------------ |
> | WebSailor       | 3.2                                 | 40.0                         | 20                 |
> | DeepDive (ours) | **5.2**                             | **32.8**                     | **3250**           |
>
> The **higher fuzzy-entity count (5.2 vs. 3.2)** and **lower R1 accuracy (32.8% vs. 40.0%)** indicate that DeepDive questions are **more challenging** than WebSailor’s and require a deeper search to answer. Additionally, we release **all 3,250 QA pairs** and the complete code for constructing the KG-based data, enabling the community to reproduce our data and extend the pipeline.

---

> > ### Author Response · Authors · 2025-11-21
> > **Response to Reviewer GkUg (2/2)**
> >
> > > **W3 & Q3: Use recent models. The experiments may also evaluate on the Qwen3-32B model, which is enhanced on tool use ability comparing the QwQ-32B.**
> >
> > Thank you for your valuable suggestion. We additionally train DeepDive on the newer Qwen3-32B backbone, re-running both SFT and RL under the same setup (same data, hyperparameters, and RL framework) to compare with QwQ-32B:
> >
> > | Base Model           | BrowseComp | BrowseComp-ZH | Xbench-DeepSearch | SEAL-0   |
> > | -------------------- | ---------- | ------------- | ----------------- | -------- |
> > | QwQ-32B (SFT only)   | 9.5        | 23.0          | 48.5              | 23.9     |
> > | Qwen3-32B (SFT only) | 11.7       | 25.7          | 54.0              | **27.7** |
> > | QwQ-32B (RL)         | **15.3**   | **29.7**      | 51.8              | 25.5     |
> > | Qwen3-32B (RL)       | 14.9       | 25.5          | **57.0**          | 25.0     |
> >
> > At the SFT stage, **Qwen3-32B clearly outperforms QwQ-32B on all four benchmarks**, which is consistent with its stronger tool-use ability. However, at the RL stage, the two 32B models **achieve very similar BrowseComp scores** (15.3 vs. 14.9), suggesting that at this scale the final RL performance may be limited by model capacity.

---

### Official Review · Reviewer_7TJL · 2025-11-02

**Soundness:** 3
**Presentation:** 3
**Contribution:** 1
**Rating:** 4
**Confidence:** 4

**Summary:**

Augmenting large language models (LLMs) with browsing tools greatly enhances their potential as deep search agents capable of tackling complex, real-world tasks. However, open-source LLMs still struggle in such settings due to limited long-horizon reasoning with browsing tools and the lack of sufficiently challenging supervised data. To overcome these limitations, this paper introduces DeepDive, a framework designed to advance deep search agents. First, it proposes an automated strategy for synthesizing complex, difficult, and hard-to-find questions from open knowledge graphs. Second, it employs end-to-end multi-turn reinforcement learning (RL) to improve LLMs’ deep search and long-horizon reasoning capabilities. Built upon open models, DeepDive-32B achieves 15.3% accuracy on BrowseComp and demonstrates strong test-time scaling in both tool usage and parallel sampling.

**Strengths:**

Clarity:
1. The paper presents its methods and experimental analyses clearly.
2. It provides sufficient ablation experiments to support the effectiveness of the proposed approach.

**Weaknesses:**

Originality: How does your work differ from “WebSailor: Navigating Super-human Reasoning for Web Agent”? I believe your approach shows little innovation in both the knowledge graph–based data construction and the multi-turn RL training methods.

**Questions:**

1. What obfuscation strategies are used in the paper?

2. Regarding the multi-turn RL training method mentioned: it seems that the agent performs multi-step tool calls to produce an answer — but isn’t this the standard approach in existing work? There are no methods that produce an answer in a single tool call, right? Or do you compute rewards separately for each intermediate tool call? Based on my understanding, your method assigns a single reward to the entire multi-step trajectory, combining answer correctness and a redundancy penalty.

3. Is your training and evaluation framework based on open-source code?

---

> ### Author Response · Authors · 2025-11-21
> **Response to Reviewer 7TJL (1/3)**
>
> Thank you very much for your constructive suggestions. We provide point-by-point responses to clarify our contributions further:
>
> > **W1: How does your work differ from “WebSailor: Navigating Super-human Reasoning for Web Agent”?**
>
> Our work differs from WebSailor in two key aspects:
>
> - **RL design**: Current search agents suffer from redundant, similar queries; therefore, we introduce a redundancy penalty for repeated queries, **reducing the average number of tool calls on BrowseComp by approximately 14%**.
> - **Data construction.** Compared to WebSailor, DeepDive uses controlled KG random walks with entity obfuscation to generate systematically more complex questions, as evidenced by a **higher number of fuzzy entities per question (5.2 vs. 3.2)** and **a lower solve rate by DeepSeek-R1 (avg@4: 32.8% vs. 40.0%)**.
>
> At the same model size (32B), DeepDive outperforms WebSailor on BrowseComp by **+32% (SFT) and +46% (RL)**; details follow below.
>
> 1. **RL Training Design (Contribution 1)**
>
> DeepDive augments the RL objective with a **redundancy-aware term** to explicitly **improve search efficiency in long-horizon, deep-search scenarios**. In practice, RL tends to encourage more and more tool calls, which may increase accuracy but also lead to redundant, overlapping queries and unnecessarily long trajectories. To counter this, DeepDive defines a trajectory-level redundancy metric and subtracts a penalty from the final reward when the agent issues multiple similar queries, thereby incentivizing concise and non-repetitive search behavior.
>
> On BrowseComp, this redundancy-aware design allows the agent to solve challenging questions with fewer unnecessary tool calls, reducing the average number of tool calls by about **14% (from 42 to 36)** while maintaining or slightly improving performance:
>
> | **Setting**         | **Redundancy Penalty** | Accuracy $\uparrow$ | **Avg. Tool Counts** $\downarrow$ |
> | ------------------- | ---------------------- | ------------------- | --------------------------------- |
> | GRPO (vanilla)      | ✗                      | 14.8                | 42                                |
> | **DeepDive (ours)** | ✓                      | **15.3**            | **36**                            |
>
> 2. **Data Construction (Contribution 2)**
>
> DeepDive constructs QA pairs from knowledge graphs using **controlled** random walks with entity obfuscation. We enforce longer paths (>5 hops) and increase the number of obfuscated entities per question, so each question contains multiple uncertain entities that require a deep search to resolve.
>
> To quantify this, we compare **all 20 released WebSailor examples** with **32 randomly sampled DeepDive examples** (from our complete set of 3,250). We prompt QwQ-32B to count fuzzy entities per question and measure the proportion of questions solved by DeepSeek-R1-0528 (averaged over 4 runs to reduce variance):
>
> | **Method**          | **Fuzzy Entity Number**  $\uparrow$ | **R1 Accuracy** $\downarrow$ | **Open Data Size** |
> | ------------------- | ----------------------------------- | ---------------------------- | ------------------ |
> | WebSailor           | 3.2                                 | 40.0                         | 20                 |
> | **DeepDive (ours)** | **5.2**                             | **32.8**                     | **3,250**           |
>
> The higher fuzzy-entity count (5.2 vs. 3.2) and lower R1 accuracy (32.8% vs. 40.0%) indicate that DeepDive queries are **more challenging**, encouraging the model to perform a deeper search.
>
> In addition, WebSailor releases only 20 samples and no data-construction method, whereas DeepDive **releases all 3,250 QA pairs and the KG-based construction pipeline**, enabling reproducible data generation and further development by the community.
>
> 3. **Supporting Results**
>
> The differences in data construction and RL design are reflected in the empirical comparison at the same model size on both BrowseComp and BrowseComp_ZH. Compared to WebSailor, **DeepDive achieves significant performance gains**, with improvements of +32% and +20% in SFT (which reflects data quality) and +46% and +16% in RL (which reflects the RL training framework):
>
> | **Model**               | **Type** | **BrowseComp**  | **BrowseComp_ZH** |
> | ----------------------- | -------- | --------------- | ----------------- |
> | WebSailor-32B           | SFT      | 7.2             | 19.2              |
> | **DeepDive-32B (ours)** | **SFT**  | **9.5 (+32%)**  | **23.0 (+20%)**   |
> | WebSailor-32B           | RL       | 10.5            | 25.5              |
> | **DeepDive-32B (ours)** | **RL**   | **15.3 (+46%)** | **29.7 (+16%)**   |
>
> Overall, DeepDive synthesizes **more challenging deep-search questions** and trains agents with **a redundancy-aware RL item** that discourages repeated or similar tool calls. These two components jointly yield consistent gains in both accuracy and search efficiency.

---

> > ### Author Response · Authors · 2025-11-21
> > **Response to Reviewer 7TJL (2/3)**
> >
> > > **Q1: What obfuscation strategies are used in the paper?**
> >
> > We design obfuscation to work with deep KG random walks so that each question contains several uncertain entities while preserving the original reasoning path:
> >
> > - **Attribute-level obfuscation.** Along each KG random-walk path, we treat the backbone as a **long chain of entities**, each with many attributes. During obfuscation, we maintain this backbone and only blur attributes that would make the answer trivial (e.g., exact years, precise locations, full names). For example, “in 1948” becomes “in the late 1940s”, and a concrete place name becomes “in the place where the event took place.”
> > - **LLM-based chain rewriting.** We use Gemini-2.5-Pro (empirically best in our manual tests, due to its strong world knowledge) to rewrite KG-derived chains into high-quality QA pairs so that identity-revealing attributes are systematically replaced with fuzzy descriptions, while preserving KG relations and the correct answer.
> >
> > These obfuscation strategies are applied to data construction, yielding questions with more fuzzy entities and more challenging QA pairs (as quantified above), providing a stronger supervision signal for RL training.
> >
> > > **Q2: Regarding the multi-turn RL training method mentioned: it seems that the agent performs multi-step tool calls to produce an answer — but isn’t this the standard approach in existing work? There are no methods that produce an answer in a single tool call, right? Or do you compute rewards separately for each intermediate tool call? Based on my understanding, your method assigns a single reward to the entire multi-step trajectory, combining answer correctness and a redundancy penalty.**
> >
> > 1. **Multi-step tool calls are just standard practice**
> >
> > Prior web-agent work typically operates in settings where 3–5 tool calls are sufficient (e.g., HotpotQA, 2WikiMultiHopQA), resulting in relatively short trajectories.
> >
> > **However, BrowseComp has a much longer horizon.** On BrowseComp, a vanilla QwQ-32B agent with standard function calling scores **1.7% with only 1.5 tool calls** per question on average, while our DeepDive-32B (after SFT + RL) reaches **15.3% with about 36 tool calls** per question.
> >
> > Our contribution enables long-horizon deep search for agents. We first use automatically synthesized, high-difficulty data to instill long-horizon search patterns in SFT, and then utilize RL to encourage deeper search further.
> >
> > 2. **Assign rewards to intermediate tool calls**
> >
> > We **do not** compute rewards for individual tool calls. As in DeepSeek-R1 and related work, we assign a reward only to the **entire multi-step trajectory**. We have two reasons for this:
> >
> > (1) For deep-search questions, there are often **many valid tool-call sequences** leading to the correct answer. Step-level rewards would bias the agent toward a specific path, **reducing applicable exploration**.
> >
> > (2) It is difficult to reliably decide whether each intermediate tool call is “good” or “bad”; process reward would introduce **noise and bias**. However, the outcome reward can avoid this problem.
> >
> > 3. **Trajectory-level reward**
> >
> > We use a single trajectory-level reward that first boosts accuracy, then explicitly addresses repeated/similar queries: Starting from a DeepDive-32B SFT model with a score of **9.5**, applying RL with a **binary correctness reward** only raises performance to **14.8** on BrowseComp; however, the agent tends to issue many redundant and overlapping tool calls (averaging **42** calls per question). We then augment this trajectory-level reward with a redundancy penalty computed from a trajectory-level metric that measures repeated or highly similar queries with little new information.
> >
> > With this redundancy-aware objective, performance is maintained or slightly improved (from **14.8 to 15.3**). At the same time, **the average number of tool calls drops by approximately 14% (from 42 to 36)**, indicating that the agent learns to avoid repeated queries.

---

> > > ### Author Response · Authors · 2025-11-21
> > > **Response to Reviewer 7TJL (3/3)**
> > >
> > > > **Q3: Is your training and evaluation framework based on open-source code?**
> > >
> > > Yes. Both our evaluation and training pipelines are built on open-source code, and we follow the standard Qwen function-call schema.
> > >
> > > 1. **Evaluation framework**
> > >
> > > **For the function-call schema,** we adopt the open-source Qwen function-call framework [1] to run our web agent. At each step, the model first produces a short reasoning trace and then emits a structured JSON tool call that specifies the tool name and its arguments.
> > >
> > > **For multi-turn search**, we use Serper and Jina as web search backends. The agent issues a sequence of such tool calls, accumulates evidence across turns, and finally returns an answer based on the collected information.
> > >
> > > **For reward computation,** we determine whether the final answer is correct using an LLM-as-judge setup adapted from the official BrowseComp script [2]. We (i) replace o3-mini with the open-source Llama-3.1-70B [3] to reduce evaluation cost, and (ii) broaden the judging prompt to accept semantically equivalent answers (e.g., paraphrases and minor formatting differences). We manually check a sample of cases and find that the new judge agrees with the original o3-mini decisions in over 95% of instances, suggesting that the automatic evaluation remains accurate.
> > >
> > > 2. **Training framework**
> > >
> > > Both SFT and RL are implemented on the open-source slime framework [4]. We reproduce the multi-search interaction used in evaluation to maintain consistency between training and evaluation, and add an early stopping mechanism to accelerate training and a redundancy penalty to reduce unnecessary tool calls.
> > >
> > >
> > >
> > > **Reference**
> > >
> > > [1] Qwen Team. Qwen function call framework. https://qwen.readthedocs.io/en/latest/framework/function_call.html
> > >
> > > [2] OpenAI. BrowseComp evaluation script (LLM-as-judge prompt). https://github.com/openai/simple-evals/blob/main/browsecomp_eval.py#L25-L45
> > >
> > > [3] Meta AI. Llama-3.1-70B model card. https://huggingface.co/meta-llama/Llama-3.1-70B
> > >
> > > [4] slime Team. slime: An LLM post-training framework for RL scaling. https://github.com/THUDM/slime

---

### Note · Authors · 2026-01-05

I have read and agree with the venue's withdrawal policy on behalf of myself and my co-authors.